# ROS-Induced DNA-Damage and Autophagy in Oral Squamous Cell Carcinoma by *Usnea barbata* Oil Extract—An In Vitro Study

**DOI:** 10.3390/ijms232314836

**Published:** 2022-11-27

**Authors:** Violeta Popovici, Adina Magdalena Musuc, Elena Matei, Oana Karampelas, Emma Adriana Ozon, Georgeta Camelia Cozaru, Verginica Schröder, Laura Bucur, Ludmila Aricov, Mihai Anastasescu, Mariana Așchie, Victoria Badea, Dumitru Lupuliasa, Cerasela Elena Gîrd

**Affiliations:** 1Department of Microbiology and Immunology, Faculty of Dental Medicine, Ovidius University of Constanta, 7 Ilarie Voronca Street, 900684 Constanta, Romania; 2“Ilie Murgulescu” Institute of Physical Chemistry, Romanian Academy, 202 Spl. Independentei, 060021 Bucharest, Romania; 3Center for Research and Development of the Morphological and Genetic Studies of Malignant Pathology, Ovidius University of Constanta, CEDMOG, 145 Tomis Blvd., 900591 Constanta, Romania; 4Department of Pharmaceutical Technology and Biopharmacy, Faculty of Pharmacy, Carol Davila University of Medicine and Pharmacy, 6 Traian Vuia Street, 020956 Bucharest, Romania; 5Clinical Service of Pathology, Sf. Apostol Andrei Emergency County Hospital, 145 Tomis Blvd., 900591 Constanta, Romania; 6Department of Cellular and Molecular Biology, Faculty of Pharmacy, Ovidius University of Constanta, 6 Capitan Al. Serbanescu Street, 900001 Constanta, Romania; 7Department of Pharmacognosy, Faculty of Pharmacy, Ovidius University of Constanta, 6 Capitan Al. Serbanescu Street, 900001 Constanta, Romania; 8Department of Pharmacognosy, Phytochemistry, and Phytotherapy, Faculty of Pharmacy, Carol Davila University of Medicine and Pharmacy, 6 Traian Vuia Street, 020956 Bucharest, Romania

**Keywords:** *Usnea barbata* (L.) Weber ex F.H. Wigg, canola oil, phenolic secondary metabolites, reactive oxygen species, cytotoxicity, CLS-354 OSCC cell line, blood cell cultures

## Abstract

Oxidative stress is associated with aging, cancers, and numerous metabolic and chronic disorders, and phenolic compounds are well known for their health-promoting role due to their free-radical scavenging activity. These phytochemicals could also exhibit pro-oxidant effects. Due to its bioactive phenolic secondary metabolites, *Usnea barbata* (L.) Weber ex. F.H. Wigg (*U. barbata*) displays anticancer and antioxidant activities and has been used as a phytomedicine for thousands of years. The present work aims to analyze the properties of *U. barbata* extract in canola oil (UBO). The UBO cytotoxicity on oral squamous cell carcinoma (OSCC) CLS-354 cell line and blood cell cultures was explored through complex flow cytometry analyses regarding apoptosis, reactive oxygen species (ROS) levels, the enzymatic activity of caspase 3/7, cell cycle, nuclear shrinkage (NS), autophagy (A), and synthesis of deoxyribonucleic acid (DNA). All these studies were concomitantly performed on canola oil (CNO) to evidence the interaction of lichen metabolites with the constituents of this green solvent used for extraction. The obtained data evidenced that UBO inhibited CLS-354 oral cancer cell proliferation through ROS generation (316.67 × 10^4^), determining higher levels of nuclear shrinkage (40.12%), cell cycle arrest in G0/G1 (92.51%; G0 is the differentiation phase, while during G1 phase occurs preparation for cell division), DNA fragmentation (2.97%), and autophagy (62.98%) than in blood cells. At a substantially higher ROS level in blood cells (5250.00 × 10^4^), the processes that lead to cell death—NS (30.05%), cell cycle arrest in G0/G1 (86.30%), DNA fragmentation (0.72%), and autophagy (39.37%)—are considerably lower than in CLS-354 oral cancer cells. Our work reveals the ROS-mediated anticancer potential of UBO through DNA damage and autophagy. Moreover, the present study suggests that UBO pharmacological potential could result from the synergism between lichen secondary metabolites and canola oil phytoconstituents.

## 1. Introduction

Reactive oxygen species display a dual role in human physiological processes [1]. Low ROS concentrations intervene in biosynthetic processes—contributing to thyroid hormone synthesis [2], cell signaling for cell growth, differentiation, proliferation, and survival [3], and host defense against various pathogens [4]. In overproduction, ROS interact with carbohydrates, lipids, proteins, and nucleic acids, leading to permanent functional changes or deterioration [5]. The cell structures which contain these organic constituents are damaged, leading to a wide range of pathologies [6]. The regulating pathways of ROS homeostasis (the steady-state control over ROS production–detoxification) are essential for diminishing ROS toxicity [7]. Oxidative stress is installed when biochemical processes leading to ROS production are over those responsible for their removal [8]. It is associated with aging, numerous metabolic and chronic disorders, and various malignancies [9]. Oral cancer is one of the most invasive neoplasia, with a high recurrence and a 5-year survival rate of 50–60% [10].

Therefore, suitable antioxidants should be able to scavenge the free radicals, forming a new radical that is stable on further oxidation [11].

As plant secondary metabolites, phenolic compounds are well known for their health-promoting role [12] due to their antioxidant potential [13]. Moreover, these phytochemicals could also exhibit pro-oxidant effects, depending on environmental conditions [14], related to the phenolic content in matrix media. The redox cycle is catalyzed by various metals; thus, formed phenolic radicals induce lipids peroxidation and DNA damage [15]. Both behaviors could be helpful in anticancer activity, being implied in prevention and therapy [16].

Phenolic metabolites also display considerable antibacterial activity, interacting with bacterial cell surfaces [17]. Generally, Gram-positive bacteria are more susceptible than Gram-negative ones due to their action. The Gram-negative bacteria’s resistance to plant phenolic compounds is due to a complex outer membrane linked to their cell wall, which hinders the phytoconstituents’ penetration [18]. The phenolics’ lipophilic character augments their antimicrobial potential [19] by facilitating their passage through the cell membrane.

Generally, plant extracts with high phenolic content show significant antioxidant and cytotoxic activities, but their correlation is not always straightforward [15]. The cytotoxicity mechanisms were commonly related to cell cycle arrest, leading to apoptosis [20].

In the plant world, lichens are unique symbionts between fungi and algae, distinguished by specific secondary metabolites [21,22,23,24] with phenolic structures. They are implied in defense against ultraviolet radiation or various pathogens’ aggression, thus assuring the lichen’s survival in difficult environmental conditions. The lichen secondary metabolites are stored in the cortex or medulla and can be found on fungal hyphae as crystals [25]; their distribution in the thallus layers correlates with bioactivities [26]. Their most significant pharmacological effects are antioxidant [27,28], antimicrobial [29,30], anticancer [31,32], photoprotective [33,34,35], and anti-inflammatory [36]. Therefore, they are considered important bioactive representatives with pharmaceutical applications [37] as a potential source of anticancer [38,39,40] and antibiotic medicines [41,42,43]. Recently, a few authors performed lichens incorporation in different pharmaceutical formulations [44,45,46,47,48,49].

The lichens belonging to the genus *Usnea* (*Parmeliaceae*) are known for their antioxidant [50,51,52], antimicrobial [53,54,55,56,57], and anticancer [58,59,60,61] effects. As a valuable representative of the *Usnea* genus, *U. barbata* has been used as phytomedicine for thousands of years [62,63,64]. Its biological potential is mainly attributed to a lipophilic phenolic compound, (+)- usnic acid [65,66,67,68]. Other minor secondary metabolites: phenolic acids [69], depsides, depsidones, and diphenyl ethers [70], could act synergistically with usnic acid in *U. barbata* extracts.

Many authors have studied the pharmacological potential of isolated metabolites and *Usnea* sp. extracts in different chemical solvents [71,72,73,74,75,76]. In 2020, Basiouni et al. [77] described the sunflower oil extract of *U. barbata*, evidencing its antibacterial and cytotoxic properties. We prepared the *U. barbata* extract in another green solvent [78,79], canola oil [80].

The present work proposes to analyze the properties of *U. barbata* extract in canola oil. First, we analyzed the influence of lichen phenolic metabolites extracted in UBO on CNO morphology during heating and its rheological properties. Then, complex flow cytometry analyses explored the UBO cytotoxicity on OSCC cell line CLS-354 and blood cell cultures. In addition, its inhibitory activity against oral pathogens implied in oral carcinogenesis [81,82] and buccal infections in immunocompromised patients [83,84,85] was evaluated. All these studies were concomitantly performed on canola oil, aiming to evidence the interaction between *Usnea* lichen metabolites and CNO phytoconstituents.

## 2. Results

### 2.1. Atomic Force Microscopy

The AFM images of both oil samples are displayed in Figure 1—UBO (Figure 1a) and, respectively, CNO (Figure 1b)—scanned over an area of (8 × 8) µm^2^. The characteristic line scans (surface profile) are plotted for both samples at the positions indicated by horizontal red lines.

The UBO exhibited a smooth surface, with low RMS roughness values: 0.55 nm at (8 × 8) μm^2^ and 0.42 nm at (3 × 3) μm^2^ (Figure 1e–h). This aspect is also reflected by the surface profile plotted below the AFM surface (Figure 1a), with most features located at a vertical height of ~2 nm. Random small agglomerations (in drop form) are visible on the surface of UBO. The peak-to-valley values registered for UBO are 11.3 nm at (8 × 8) μm^2^ and 8.2 nm at (3 × 3) μm^2^.

On the other hand, CNO shows a significant accumulation of materials (Figure 1b), probably due to a stronger tendency towards polymerization [86]. These ones, which have 10–20 nm in height, lead to an increase in RMS roughness, as can be seen in Figure 1e–h: 4.73 nm at (8 × 8) μm^2^ and 1.99 nm at (3 × 3) μm^2^. The peak-to-valley parameter of CNO is ~65 nm at (8 × 8) μm^2^ and 16.7 nm at (3 × 3) μm^2^.

### 2.2. Rheology

Steady shear measurements were achieved at 25 °C to determine the flow behavior of CNO and UBO. As is shown in Figure 2a, a direct proportionality between both moduli and frequency is evidenced. The energy stored in the oil sample is the storage modulus (G’). The loss modulus (G’’) represents the energy lost during deformation [87]. Both oil samples’ storage modulus (G’) values are near zero, showing liquids with no elastic properties, where all stored energy is scattered as heat [87]. The viscous modulus (G”) is more significant than G’, which means that the energy requested to deform the samples is viscously dispersed, a liquid-like behavior. It is a characteristic of vegetable oils with unsaturated fatty acids [88].

Storage and loss moduli (as a function of frequency) are represented in Figure 2b for CNO and UBO. Loss modulus (G”) values slightly increased with frequency. In contrast, the storage modulus (G’) showed a higher increase with frequency. At low frequencies, the G” value was substantially higher than the G’ one. No differences were observed between CNO and UBO moduli values. The UBO’s storage modulus at a lower frequency is slightly higher than CNO. When the frequency increases, G’ quickly elevates compared to G” until G’ become higher than G” after the cross-point, indicating a loss in the liquid-like behavior for both oil samples.

The shear viscosities of the CNO and UBO diminished when the shear rate augmented and achieved a near-constant value (Figure 2c). When a shearing force is applied, the shear-thinning behavior is caused by the reversible alteration of the oils’ well-ordered structure. When this process was complete, no supplementary reduced viscosity was detected because the shear rate was increased. Figure 2c shows that the CNO and UBO viscosities depend on the shear rates, especially for values below 10^−1^ s^−1^, indicating a non-Newtonian behavior. When shear stress exceeds 10^−1^, the viscosity remains constant, exhibiting a Newtonian flow.

### 2.3. Annexin V-FITC Apoptosis Assay

Lipids are asymmetrically distributed in viable cells on the plasma membrane’s inner and outer leaflets. Phosphatidylserine (PS) is one of these lipids, generally exposed to the cell cytoplasm. During apoptosis, PS is externalized, becomes detectable for FITC, and Annexin V binds to it [89]; PI cannot cross healthy membranes—it only can enter the cells through altered ones. Propidium iodide stains the nucleic acids inside dead cells or those with reversible membrane destruction (late apoptotic and necrotic cells) [90]. The results are illustrated in Figure 3 and Appendix A.

UBO [3:3] diminished the viability of blood cells compared to CNO [3:3] and C1 and C2P controls: 72.10 ± 1.98 vs. CNO [3:3]: 93.61 ± 1.19; C1: 99.48 ± 0.70; C2P: 99.47 ± 0.52, *p* < 0.01 (Figure 3a,g,m,n and Appendix A). Still, C3UA induced the lowest blood cell viability (61.43 ± 0.88 vs. 72.10 ± 1.98, *p* < 0.01). UBO [3:3] triggered minimal EA compared to C3UA (positive control): 0.18 ± 0.15 vs. 37.04 ± 0.66, *p* < 0.01). The lower UBO concentrations [3:2] and [3:1] did not induce EA in blood cells; then, their viability remained significantly higher compared to C3UA: 87.76 ± 2.12; 92.80 ± 0.54 vs. 61.43 ± 0.88, *p* < 0.01 (Figure 3e,f and Appendix A).

In CLS-354 tumor cells, after 24 h exposure to all UBO concentrations, the viability decreased compared to C1 (negative control): 93.61 ± 1.19; 97.83 ± 0.61; 97.55 ± 1.16 vs. C1:99.39 ± 0.75, *p* < 0.05; *p* < 0.01; *p* ≥ 0.05 (Figure 3d–f,m and Appendix A). However, V remained significantly increased compared to both positive controls: C2P: 72.51 ± 2.51, C3UA: 54.05 ± 1.68, *p* < 0.01 (Figure 3d–f,q,r,s). It can also be observed that any UBO concentration did not induce early apoptosis in tumor cells (Figure 3d–f,r and Appendix A).

After staining with Annexin V FITC/PI, in FM images (Figure 3s,t), the viable cells (unstained) and early apoptotic ones, in a green color, (Figure 3s) could be differentiated from late apoptotic cells mixed with necrotic ones—green membrane with dark orange fragmented nuclei (Figure 3t).

### 2.4. Total ROS Activity Assay

The results are illustrated in Figure 4 and Appendix A. In normal blood cells, all UBO concentrations determined an intense ROS generation compared to all controls: 5250.00 × 10^4^ ± 50.00; 5033.00 × 10^4^ ± 57.73; 3733.00 × 10^4^ ± 57.73 vs. C1:242.00 × 10^4^ ± 2.00; C2P: 311.00 × 10^4^ ± 9.64; C3UA: 846.66 × 10^4^ ± 5.77, *p* < 0.01 (Figure 4a–c,n–o). ROS levels induced by the UBO of [3:2] and [3:1] had substantially higher values compared to CNO of the same concentrations: 5033.00 × 10^4^ ± 57.73 vs. 3366.67 × 10^4^ ± 57.74; 3733.00 × 10^4^ ± 57.74 vs. 2716.67 × 10^4^ ± 28.87, *p* < 0.01 (Figure 4b,c,h,i and Appendix A).

In CLS-354 tumor cells, UBO significantly stimulated ROS production compared to 1% DMSO and 5% P407: 316.67 × 10^4^ ± 28.87; 200.00 × 10^4^ ± 20.00; 90.00 × 10^4^ ±17.32 vs. C1: 15.67 × 10^4^ ± 4.04, C2P: 96.67 × 10^4^ ± 20.82; *p* < 0.01 (Figure 4d–f,p and Appendix A). However, usnic acid (C3UA positive control) induced the highest ROS level in OSCC cells: 966.67 × 10^4^ ± 57.74, *p* < 0.01 (Figure 4d,r and Appendix A). Moreover, UBO of [3:3 and 3:2] generated considerably higher ROS levels than CNO ones: 316.67 × 10^4^ ± 28.87; 200.00 × 10^4^ ± 20.00 vs. 96.67 × 10^4^ ± 5.77; 46.67 × 10^4^ ± 5.77, *p* < 0.01 (Figure 4d,e,j,k and Appendix A).

### 2.5. Enzymatic Activity of Caspase 3/7

Both effector caspases combine their roles in cell death [91], and the obtained data are shown in Figure 5 and Appendix A.

In blood cells, the UBO of [3:3 and 3:2] considerably activated caspase is 3/7 compared to 1% DMSO (43.82 ± 5.28; 37.10 ± 0.45; vs. C1:29.26 ± 1.97, *p* < 0.05); still, their activity is significantly lower compared to C3UA: 53.98 ± 0.27, *p* < 0.01 (Figure 5a–c,m,o and Appendix A).

In CLS-354 tumor cells, UBO caspase activation was significantly higher compared to CNO (36.89 ± 1.44 vs. 24.21 ± 0.66, *p* < 0.01; 26.64 ± 0.85 vs. 22.90 ± 1.94, *p* < 0.05; 20.56 ± 0.76 vs. 16.80 ± 0.97, *p* < 0.01 (Figure 5a–i and Appendix A). The UBO of [3:3] stimulated the enzymatic activity of caspase 3/7 more than both positive controls (36.89 ± 1.44 vs. C2P: 9.60 ± 0.75, *p* < 0.01; C3UA: 27.02 ± 1.64, *p* < 0.05 (Figure 5d,r and Appendix A). Lower UBO concentrations [3:2 and 3:1] induced caspase 3/7 activation significantly higher than P407: 26.64 ± 0.85; 20.56 ± 0.76 vs. 9.60 ± 0.75, *p* < 0.01 (Figure 5d,e,r and Appendix A).

### 2.6. Cell Cycle Assay

The cell cycle profile was determined by staining the cellular DNA with PI, allowing differentiation of cells in subG0/G1, G0/G1, S phase, and G2/M. The results are displayed in Figure 6 and Appendix A.

In blood cells, all three concentrations of UBO and CNO blocked DNA synthesis, while the controls recorded low DNA synthesis values: 0.00 ± 0.00 vs. C2P:1.79 ± 0.36, *p* < 0.05; C1:4.76 ± 0.68, *p* < 0.01; C3UA: 2.86 ± 0.23, *p* < 0.01 (Figure 6a–c,m–o and Appendix A). All concentrations of UBO also induced cell cycle arrest in G0/G1 (86.30 ± 0.53; 82.67 ± 2.43; 72.86 ± 4.06); UBO of [3:3 and 3:2] did not report substantial differences compared to CNO [3:3 and 3:2] and controls; CNO: 91.64 ± 1.52, *p* < 0.05, 89.66 ± 1.30, *p* < 0.05, 65.70 ± 0.73, *p* ≥ 0.05; C1: 88.52 ± 0.74; *p* ≥ 0.05; *p* < 0.01; *p* < 0.05, C2P: 85.35 ± 4.94; *p* ≥ 0.05; *p* ≥ 0.05; *p* < 0.05, C3UA: 90.05 ± 3.45; *p* ≥ 0.05; *p* < 0.05; *p* < 0.01 (Figure 6a–c,m–o and Appendix A).

In CLS-354 tumor cells, all UBO dilutions [3:3, 3:2, and 3:1] determined cell cycle arrest in G0/G1 phase (92.51 ± 0.62; 87.17 ± 1.57; 74.37 ± 1.27 vs. CNO: 91.31 ± 0.84, *p* ≥ 0.05, 90.69 ± 0.94, *p* < 0.05, 60.96 ± 0.39, *p* < 0.01; C1: 92.13 ± 1.61, *p* ≥ 0.05, *p* < 0.01, *p* < 0.01; C2P: 83.56 ± 2.46, *p* ≥ 0.05, C3UA: 90.05 ± 3.45, *p* ≥ 0.05, *p* ≥ 0.05, *p* < 0.05 (Figure 6d–f,p,r and Appendix A). A lower DNA synthesis was observed reported to 5% P407 (4.90 ± 0.53; 6.56 ± 0.48; 4.08 ± 1.46 vs. 8.26 ± 1.64, *p* < 0.05, *p* ≥ 0.05, *p* < 0.05 (Figure 6d–f,r and Appendix A). The lowest concentration of UBO [3:1] determinates a cell arrest in G2/M, also reported to CNO [3:1] and controls: 19.20 ± 2.22 vs. 28.39 ± 2.07, *p* < 0.05; C1:1.05 ± 0.27, *p* < 0.01; C2P: 5.12 ± 2.19, *p* < 0.05; C3UA: 4.06 ± 1.45, *p* < 0.05.

### 2.7. Nuclear Shrinkage and Autophagy

The results are displayed in Figure 7, Figure 8, Appendix A. Pyknotic nuclei stained with Hoechst 33342 in blood cells and CLS-354 tumor cells were indicated by flow cytometry in Figure 7 and Appendix A.

In blood cells, UBO [3:3; 3:2; 3:1] determines a significant increase in nuclear shrinkage compared to CNO and UA: 30.05 ± 1.66 vs. 21.59 ± 1.46, *p* < 0.05; 22.17 ± 2.43 vs. 15.68 ± 1.63, *p* ≥ 0.05; 15.26 ± 0.78 vs. 6.63 ± 0.29, *p* < 0.01; vs. C3UA: 3.19 ± 0.30, *p* < 0.01 (Figure 7a–c,g–i,m–o and Appendix A).

In CLS-354 tumor cells, the UBO-induced nuclear shrinkage registered higher levels than in normal blood ones, with considerable differences compared to 1% DMSO and P407: 40.12 ± 1.35 vs. 31.29 ± 0.95, *p* < 0.05; 31.71 ± 0.96 vs. 23.86 ± 1.97, *p* < 0.01; 25.27 ± 1.26 vs. 13.74 ± 0.42, *p* < 0.05; vs. C1:16.11 ± 3.11, *p* < 0.01; *p* < 0.01; *p* ≥ 0.05; C2P: 20.06 ± 0.37, *p* < 0.05, *p* < 0.01; *p* < 0.01 (Figure 7d–f,j–l,p–r and Appendix A).

The cells were stained with acridine orange [92] to show intensified autophagy by oxidative stress after UBO treatment (Figure 8).

In blood cells treated with UBO [3:3; 3:2; 3:1], the lysosomal activity levels were significantly higher compared to CNO and UA (39.37 ± 1.62 vs. 29.87 ± 1.46, *p* < 0.01; 36.48 ± 0.72 vs. 21.18 ± 1.66, *p* < 0.01; 32.38 ± 1.86 vs. 15.71 ± 1.96, *p* < 0.05; vs. C3UA: 3.19 ± 0.30, *p* < 0.01; *p* < 0.01; *p* < 0.05. However, they registered considerably lower values than 1% DMSO (negative control) and 5% P407: C1: 51.30 ± 3.25, *p* < 0.05; C2P: 53.23 ± 1.99, *p* < 0.05; *p* < 0.01; *p* < 0.01 (Figure 8a–c,g–i,m–o and Appendix A).

Autophagy was substantially augmented in CLS-354 tumor cells after 24 h exposure to UBO compared to CNO, 1% DMSO (negative control), and 5% P407: 62.98 ± 2.06 vs. 43.90 ± 0.60, *p* < 0.01; 51.16 ± 0.69 vs. 36.50 ± 1.23, *p* < 0.01; 47.37 ± 1.18 vs. 31.17 ± 0.79, *p* < 0.01; vs. C1: 12.57 ± 0.92, *p* < 0.01; C2P: 27.27 ± 1.37, *p* < 0.01 (Figure 8d–f,j–l,p and Appendix A).

### 2.8. DNA Synthesis Assay

The effects of UBO on DNA synthesis and fragmentation in blood cells and CLS-354 tumor cells were observed through flow cytometry analyses after EdU incorporation; EdU is a nucleoside analog of thymidine. It is incorporated into DNA during active DNA synthesis [93] (Figure 9 and Appendix A). Discontinuous fragmentation of nuclear DNA during apoptosis is revealed by discrete “Sub-G1” peaks on DNA content histograms [94]. Cells accumulated in the sub-G0/G1 phase are classified as apoptotic cells with DNA fragmentation and mitochondrial dysfunction [95].

In blood cells, UBO [3:3; 3:2; 3:1] blocked DNA synthesis, reporting considerable differences compared to all controls: 0.00 ± 0.00 vs. C1:10.36 ± 1.21, *p* < 0.01; C2P: 2.75 ± 1.34, *p* ≥ 0.05; C3UA: 6.49 ± 1.25, *p* < 0.05 (Figure 9a–c,m–o,s,u,w). On the other hand, UBO-induced DNA fragmentation was significantly lower values compared to 1% DMSO and 5% P407: 0.72 ± 0.23; 1.47 ± 0.75; 0.65 ± 0.21 vs. C1: 2.01 ± 0.20; *p* < 0.01; C2P: 3.47 ± 0.79, *p* < 0.05 (Figure 9a–c,m,n and Appendix A).

In CLS-354 tumor cells, UBO [3:3; 3:2; 3:1] considerably diminished DNA synthesis compared to 1% DMSO (2.80 ± 0.63; 3.74 ± 0.34; 3.52 ± 1.02 vs. 12.44 ± 2.80, *p* < 0.05). The previously mentioned values are higher compared to 5% P407 (positive control): 1.16 ± 1.07, *p* ≥ 0.05; *p* < 0.05; *p* < 0.05 (Figure 9d–f,p,r). As shown in Figure 9d–f,p and Appendix A after UBO treatment, DNA fragmentation registered significantly lower values compared to C1 (negative control): 2.97 ± 0.84; 2.34 ± 0.39; 2.05 ± 0.66, vs. 15.18 ± 2.17, *p* < 0.05; *p* < 0.05; *p* < 0.01.

### 2.9. Principal Component Analysis

The Principal Component Analysis (PCA, Figure 8a–d) was performed for UBO, CNO, and all controls, regarding the variable parameters examined in both cell types.

In Appendix A, the correlation matrix and the PCA-Correlation circle were assessed using XLSTAT v. 2022.2.1.1309. Moreover, Agglomerative Hierarchical Clustering (AHC) Dendrogram was obtained using Euclidean distance (Appendix A).

Figure 10a,b show the places of oil samples and all controls in relationship with their activities on both cell types. Figure 10c,d compares the effects of oil samples in CLS-354 tumor cells and blood cells.

Figure 10a shows the PCA-Correlation biplot for blood cells. The two principal components explain 65.51% of total data variance, with 36.86% attributed to the first (PC1) and 28.65% to the second (PC2). The PC1 is associated with C3UA (positive control), viability, EA and LA, and subG0/G1. The PC2 is associated with C1DMSO and C2P407, oil samples (UBO [3:3], UBO [3:2], and CNO [3:3]), ROS, and necrosis. The ROS levels are lowly correlated with the enzymatic activity of caspase 3/7 (*r* = 0.226, *p* > 0.05) but significantly positively associated with necrosis (*r* = 0.789, *p* < 0.05) and negatively with DNA synthesis (*r* = −0.790, *p* < 0.05). Enzymatic activity of caspase 3/7 is moderately correlated with a cell cycle arrest in G0/G1 and LA (*r* = 0.662, *r* = 0.550, *p* > 0.05).

Figure 10b displays a PCA-Correlation biplot for CLS-354 tumor cells. The two principal components, PC1 (49.69%) and PC2 (22.86%), explained 72.54% of the total data variance. The PC1 is associated with CNO [3:2 and 3:1], C3UA, ROS, nuclear shrinkage, Viability, necrosis, EA, and LA; PC2 is associated with UBO [3:3 and 3:2], CNO3:3, C2P407, C3/7 activity, and autophagy.

The C3/7 enzymatic activity is significantly correlated with nuclear shrinkage and autophagy (*r* = 0.750, *r* = 0.711, *p* < 0.05) and is lowlily associated with ROS (*r* = 0.414, *p* > 0.05). ROS considerably correlates with LA, EA, and NS (*r* = 0.910, *r* = 0.848, *r* = 0.791, *p* < 0.05); cellular oxidative stress also moderately correlates with necrosis and autophagy (*r* = 0.604, *r* = 0.541, *p* > 0.05).

In Figure 10c,d, the two principal components (PC1 and PC2) explain 84.91% and, respectively, 87.01% of the total data variance (in blood cells, DNA synthesis is totally blocked (S = 0) and in CLS-354 ones all oil samples did not induce API.

In both cell types, ROS levels show a considerably high correlation with the enzymatic activity of caspase 3/7 (*r* = 0.988, *r* = 0.937, *p* < 0.05).

In CLS-354 cells, both mechanisms (cellular oxidative stress and caspase 3/7 enzymatic pathway) are substantially correlated with late apoptosis (*r* = 0.847, *r* = 0.873, *p* < 0.05), necrosis (*r* = 0.907, *r* = 0.940, *p* < 0.05), and autophagy (*r* = 0.941, *r* = 0.900, *p* < 0.05). Moreover, they show a strong correlation with nuclear shrinkage (*r* = 0.862, *r* = 0.929, *p* < 0.05), which is appreciably associated with a cell cycle arrest in G0/G1 (*r* = 0.823, *p* < 0.05).

In blood cells, both mechanisms are noticeably correlated with NS (*r* = 0.930, *r* = 0.957, *p* < 0.05) and A (*r* = 0.879, *r* = 0.877, *p* < 0.05), and NS is highly associated with necrosis (*r* = 0.899, *p* < 0.05).

The activity of all samples and controls on both cell types is evidenced in Appendix A. Thus, UBO [3:2] acts similarly to CNO [3:3] and UBO [3:1] with CNO [3:2]. Both pairs are followed by UBO [3:3], CNO [3:1], and C2P407, and, finally, by 1% DMSO. Only usnic acid exhibited different actions.

### 2.10. Antibacterial and Antifungal Activities

Data registered in the first level of Table 1 show the microdilutions [96] of standard antibiotic (CTR), antifungal (TRF) [46], P407 [44], and oil samples (UBO and CNO). All CTR microdilutions are over minimum inhibitory concentrations (MICs) against *S. aureus* and *P. aeruginosa*; MIC is the minimal concentration of an antimicrobial agent that will inhibit the visible growth of a microorganism after 24 h of incubation [97]. According to Phe et al. [98], regarding *S. aureus*, the MIC of CTR is 4–8 µg/mL, equal to [0.004–0.008] mg/mL. On *P. aeruginosa*, MIC is substantially higher, varying from 16 µg/mL to over 256 µg/mL (for substantially resistant strains) [99].

Table 1 shows that UBO and CNO of [15.167–7.583] mg/mL and P407 of [2.506–1.253] mg/mL have similar effects with CTR of [0.047–0.023] mg/mL on *S. aureus.* Their inhibitory activity on *P. aeruginosa* strains’ growth is higher than on *S. aureus*. Thus, UBO and CNO of [15.157–3.790] mg/mL and P407 of [2.506–0.626] mg/mL inhibit *P. aeruginosa* similarly with CTR of [1.603–0.755] mg/mL. The following concentrations [1.895–0.473] mg/mL of UBO and CNO and [0.317–0.178] mg/mL of P407 similarly act with CTR of [0.094–0.023] mg/mL. CNO of 0.237 mg/mL shows a lower activity than UBO at the same concentration and P407 of [0.078–0.039] mg/mL (Table 1).

Table 1 also shows that antifungal activity decreases in order: P407, UBO, and CNO. All samples and P407 (positive control) display a significantly higher inhibitory activity on *C. albicans* than on *C. parapsilosis*.

Thus, on *C. albicans*, P407 (in the entire range of microdilutions) has a fungicide effect, such as TRF and UBO and CNO of [15.167–7.583] mg/mL. At [3.790–1.895] mg/mL, UBO induces *C. albicans* strains low proliferation, while CNO determines a moderate one. In the last range [0.473–0.237] mg/mL, both oil samples similarly act, moderately inhibiting *C. albicans* proliferation.

On *C. parapsilosis*, P407 of [2.506–0.078] mg/mL has a fungicidal effect like TRF. A partial death of *C. parapsilosis* causes UBO of [15.167–7.583] mg/mL, while CNO exhibits only a fungistatic effect (fungal cells are alive but do not proliferate). 0.039 mg/mL P407 and UBO of [3.790–1.895] mg/mL induce a low proliferation, while CNO of [3.790–1.895] mg/mL and 0.947 mg/mL UBO similarly act, determining a low to moderate one. Finally, CNO of [0.947–0.237] mg/mL and UBO of [0.473–0.237] mg/mL induce a moderate proliferation of *C. parapsilosis* strains.

## 3. Discussion

The pharmaceutical industry uses vegetable oils to extract phenolic compounds and carotenoids and formulate drug delivery systems and biopolymers [102]. In the present study, the solvent for the extraction of *U. barbata* secondary metabolites was canola oil (CNO). Unlike chemical solvents, CNO has its own bioactive constituents [103] and could make a suitable extraction of lichen secondary metabolites. Canola oil consists of cold-pressed rapeseed (*Brassica napus*) oil with a low erucic acid content and glucosinolate one [104]. The CNO’s major constituents are triacylglycerols (97–99%); other phytocompounds: polyphenols, carotenoids, phytosterols, chlorophylls, tocopherols, monoglycerides, diglycerides, free fatty acids (FFA), and phospholipids are in a minor amount (1–3%) [105].

Canola oil is rich in monounsaturated fatty acids (MUFAs > 68%) and polyunsaturated ones (PUFA). It also is distinguished by remarkable contents of oleic acid (> 63%), γ-linolenic acid, and linoleic acid [106], and minor ones of erucic acid, palmitoleic acid, behenic acid, stearic acid, arachidic acid, eicosenoic acid [106]. CNO’s concentration of saturated fatty acids (SFA ≅ 7%) is lower than soybean and sunflower oils [107]. The bioactive constituents of canola oil are phenolic metabolites: tocopherols, carotenoids, and phytosterols [106]. Phenolic acids are mainly phenolic compounds in CNO: sinapic acid, ferulic acid, p-coumaric acid, cinnamic acid, 4-hydroxybenzoic acid, syringic acid, and vanillic acid [108]. Tocopherols of canola oil are γ- and α-tocopherol (as the significant tocopherols), while δ-, β-tocopherol, and plastochromanol-8 (PC-8) [109] are the minor ones [110]. The main phytosterols of CNO are brassicasterol (specific for rapeseed oil), stigmasterol, cholesterol, β-sitosterol, campesterol, and Δ5 -avenasterol [110]. Finally, the CNO’s carotenoids are lutein, β-carotene, and zeaxanthin [105]. The hydroxyl group (–OH) of all phenolic compounds can scavenge free radicals [106]; therefore, CNO has a considerable antioxidant effect.

A specific bioactive constituent in CNO is canolol [111], a phenolic compound formed in rapeseeds by sinapic acid decarboxylation during the conditioning step of oil production induced by heating [112]. Canolol proved to have significant bioactivities: antioxidant, anticancer, and antimutagenic [110]. Depending on the time and temperature, rapeseed oils extracted after this phase could have over 760 mg/kg of canolol [113]. A substantial amount of canolol is lost during the rapeseed oil refining process. However, concomitantly with canolol loss, a new phenolic compound is synthesized, identified as (4,6-dimethoxy-5-hydroxy-1-methyl3-(30,50-dimethoxy-40-hydroxyphenyl) indane and surnamed canolol dimer [114]. It is present in edible rapeseed oils over 63 mg/kg. Canolol-monomer and dimer showed an anti-proliferative effect on cancer cell lines (HeLa and MCF7) [115]; furthermore, canolol dimer exhibited an antioxidant potential two times higher than its monomer.

The total phenolic and usnic acid content and antiradical properties of UBO vs. CNO are displayed in Appendix A.

First, we analyzed the influence of lichen phenolic metabolites extracted in UBO on CNO physicochemical properties. Therefore, the morphological changes during heating and their rheological properties were examined.

When vegetable oils are heated at a high temperature, various chemical reactions (hydrolysis, oxidation, and polymerization) occur [116]. Previous studies proved that the natural antioxidants from herbs and spices could improve their oxidative stability [117,118]. Our AFM analysis suggested that the secondary phenolic metabolites from *U. barbata* extracted in UBO exhibited an antioxidant effect, significantly diminishing the oxidative processes triggered in canola oil by heating it to 200 °C. The AFM images of UBO and CNO (Section 2.1) support the previous observation. Our previous study showed strong correlations between DPPH free radical scavenging activity and total phenolic and usnic acid contents [119]; thus, they could break one of the phases of initiation or propagation of lipids autooxidation by hydrogen donating or electron transfer [120,121].

On the other hand, flow is a natural factor in oil production and refining technology [122]; because it also has an essential role in pharmaceutical formulation [123], we evaluated the rheology of both oil samples. Rheological properties are crucial pharmacotechnical [124] parameters that must be determined when considering UBO as a candidate for a potential human-using pharmaceutical form [125]. They provide essential information on the suitability of the manufacturing process and the selection of the types and amounts of ingredients used, influencing the final product’s in vivo behavior and performance.

The unsaturation level (polyunsaturated and monounsaturated fatty acids), the chain length, position of the OH group of the fatty acids determines the oils’ viscosity characteristics; their viscosity decreases due to reduced interactions between unsaturated fatty acid molecules [126]. This parameter varies directly proportional to PUFA content; oils viscosity decreases when the % MUFA increases [87]. Similar results were obtained for different vegetable oils [87,127]. No significant differences were detected between CNO and UBO; therefore, the *U. barbata* secondary metabolites extracted in UBO did not influence the oil viscosity.

The dual redox behavior of usnic acid [128] and other phenolic secondary metabolites underline the UBO cytotoxicity through pro-oxidant activity. Therefore, the ROS level and caspase 3/7 enzymatic activity in both cell types were measured, knowing that ROS may onset different cell death mechanisms (such as apoptosis, ferroptosis, autophagy, and necroptosis) and crosstalk between them [129]. Under oxidative stress, the mitochondrial apoptotic pathway activates cysteine-dependent aspartate-specific proteases (caspases) [130]. Caspase 3 is responsible for DNA fragmentation and morphological changes of apoptosis. In contrast, caspase 7 appears to have a more significant role in cellular viability loss; thus, the intricate function of both effector caspases is crucial in this type of cell death [131]. On the other hand, Bessadotir et al. [132] reported that proton-shuttling secondary metabolite usnic acid affects both mitochondrial and lysosomal activities in tumor cells. Apoptosis and autophagy interrelate in substantial crosstalk; therefore, caspases can recognize and cleave many autophagy-related proteins (ATGs) [133,134]. By shrinking key AGTs, caspases could enhance apoptosis, promoting the release of proapoptotic factors from mitochondria [134]. Moreover, caspases can induce autophagy in certain conditions [135,136]. Autophagy is an essential process for body homeostasis and can be stimulated by oxidative stress [137]. The final phase, with functional multivesicular bodies (MVBs) formation, confirms this process’s efficacy [138] and is required in various degenerative pathologies [139]. These acidic vesicular organelles were labeled with AO and quantified through flow cytometry [92,140].

Data analysis shows that, in blood cells, UBO [3:3 and 3:2] and CNO [3:3] induced the highest cellular oxidative stress (ROS level > 5000 × 10^4^), inhibiting DNA synthesis and diminishing cell viability through necrosis (27.49%, 12.22%, and 9.09%). These processes were induced through penetration into cells of a large amount of emulsified lipids rich in FFA from canola oil [141] associated with usnic acid and other lichen secondary metabolites in UBO. The FFA are strong pro-oxidants in O/W emulsions [142], and cellular lipotoxicity phenomena occur through FFA-induced high oxidative stress, triggering the necrotic mitochondrial pathway and leading to blood cell death. Other previous studies also described this process [143,144,145,146]. Moreover, lipid peroxidation products can adduct specific mitochondrial and autophagy-related proteins driving cellular dysfunction in an autophagic cell death way [147]. Usnic acid from UBO, as a fat-burner [148], augments ROS levels and intensifies cellular damage [149]. In the case of UBO [3:3], it can be observed that, in addition to the high percentage of necrosis, cell death also occurred through apoptosis (EA and LA). Our results prove that usnic acid from UBO intervenes in cellular shrinkage; under similar oxidative stress, CNO [3:3] induced only 9.09% necrosis (V = 91.04%); UBO [3:3 and 3:2] reported cell viability of 72.11% and, respectively, 87.76%. UBO [3:1] considerably diminishes oxidative stress (ROS = 3733.33 × 10^4^) and increases cell viability to 92.8%. The mitochondrial pathway implies caspase 3/7 activation and associated processes: total blocking of DNA synthesis, NS, and cell cycle arrest in G0/G1. In addition, autophagy occurs through cleaving AGTs, as previously described.

High ROS levels determine chromatin dysfunction, such as DNA fragmentation [130], driving cell death through apoptosis [150,151], or necrosis [152,153]. ROS-mediated DNA fragmentation is triggered and enhanced by PUFAs or their hydroperoxides through lipid peroxidation [154]. The DNA fragmentation factor (DFF) [155] is a heterodimeric protein formed of two DNA fragmentation factors 45 and 40; it can trigger DNA fragmentation in the presence of an activated caspase 3. Upon apoptosis activation, DFF45 is split by caspases 3 and 7 [152] and dissociates from DFF40. Therefore, DFF40—or caspase-activated DNase or nuclease (CAD or CPAN)—is available for DNA fragmentation into oligonucleosome-size particles, conducing towards cell death [156,157].

All cancer cells have significant ROS levels, promoting tumor development and progression [158]. However, substantial ROS production in tumor cells alters cellular metabolic processes by non-specifically damaging proteins, lipids, and DNA. Our results show that even if ROS concentration and caspase 3/7 enzymatic activity are lower than in blood cells, the UBO-induced oxidative stress in the CLS-354 cell line determines higher levels of nuclear shrinkage, cell cycle arrest in G0/G1, DNA fragmentation, and autophagy compared to blood cells. This fact suggests that phenolic secondary metabolites are implied in oxidative stress, generating ROS through pro-oxidant effects. On the other hand, in blood cells, at a ROS level of 10–20 times higher, the processes that lead to cell death—NC, cell cycle arrest in G0/G1, DNA fragmentation, and autophagy—are significantly diminished than in CLS-354 oral cancer cells. This observation suggests that UBO’s phenolic secondary metabolites could have a protective activity, reducing a substantial part of ROS, thus diminishing the processes that lead to blood cell death.

In addition, the following in vitro analyses evidenced the interaction between lichen phenolic compounds and the phytoconstituents of the vegetable oil used for green extraction [159]. The pro-oxidant effects of lichen phenolic metabolites could explain their antimicrobial effects. Phenolic acids from CNO are also known for their antibacterial activity [160]. Two of them (*p*-coumaric acid and cinnamic acid) are also found in *U. barbata* [69,72]. The UBO antibacterial and antifungal effects are higher than CNO due to *U. barbata* secondary phenolic metabolites that synergistically act with canola oil ones. Antonenko et al. [161] demonstrated that usnic acid has an uncoupling action involving calcium ions and causes dissipation of membrane potential in bacterial cells and isolated liver mitochondria. Its protonophoric activity significantly contributes to antibacterial and cytotoxic action [162]. Peralta et al. [163] also reported that oxidative and nitrosative stress mediate the usnic acid antifungal effect, significantly accumulating intracellular and extracellular ROS.

## 4. Materials and Methods

### 4.1. Materials

The chemicals used in this study (reagents and standards) had analytical purity. The usnic acid standard (98.1%), Propidium Iodide (PI), DMSO, Poloxamer 407, and Antibiotics mix solution—100 µL/mL with 10,000 U Penicillin, 10 mg Streptomycin, 25 µg Amphotericin B per 1 mL—were purchased from Sigma-Aldrich Chemie GmbH (Schnelldorf, Germany). The flow cytometry staining buffer (FCB) and Annexin V Apoptosis Detection Kit were provided by eBioscience (San Diego, CA, USA). Promega Corporation (Madison, WI, USA) produced 4 mg/mL RNase A. The following kits: EdU i-Fluor 488, Reactive Oxygen Species Detection Assay Kit, and Magic Red^®^ Caspase-3/7 Assay Kit were supplied by Abcam Plc (Cambridge, UK) [46].

Bacterial and fungal cell lines (*P. aeruginosa* ATCC 27353, *S. aureus* ATCC 25923, *C. parapsilosis* ATCC 22019, and *C. albicans* ATCC 10231) were provided by Microbiology Department, S.C. Synevo Romania S.R.L. from Constanta in the partnership agreement (No 1060/25.01.2018) with the Ovidius University of Constanta, Faculty of Pharmacy. Resazurin solution (part from In Vitro Toxicology Assay Kit, TOX8-1KT, Resazurin based) and RPMI 1640 Medium were furnished by Sigma-Aldrich Chemie GmbH (Schnelldorf, Germany). Thermo Fisher Scientific GmbH (Karlsruhe, Germany) afforded Mueller–Hinton agar (MHA) culture medium [44].

CLS Cell Lines Service GmbH (Eppelheim, Germany) was the purveyor of the CLS-354 (Human mouth squamous cell carcinoma from a 51-year-old male) growing culture and culture medium—Dulbecco’s Modified Eagle’s Medium (DMEM) High Glucose, supplemented with 10% Fetal Bovine Serum (FBS) 4.5 g/L glucose and L-glutamine. Thermo Fisher Scientific Inc. (Waltham, MA, USA) for the media for blood cells (Dulbecco’s phosphate buffered saline with MgCl_2_ and CaCl_2_), FBS, Trypsin-ethylenediamine tetra acetic acid (Trypsin EDTA), and L-Glutamine (200 mM) [48].

A non-smoker healthy donor—with B Rh+ blood type—gave the blood samples, according to Donor Consent code 39/30.06.2021 and Ovidius University of Constanta Ethical approval code 7080/10.06.2021 [44,164].

*U. barbata* was harvested in March 2021 from the Călimani Mountains—900 m altitude and the following coordinates: 47°29′ N, 25°12′ E [46]. The Department of Pharmaceutical Botany (Faculty of Pharmacy, Ovidius University of Constanta) identified it through standard methods. A voucher specimen is preserved in the Herbarium of the Pharmacognosy Department (the Ovidius University of Constanta, Faculty of Pharmacy). The producer TAF PRESSOIL SRL (Cluj, Romania) provided the canola seed oil [44,165].

### 4.2. Lichen Extract in Canola Oil

The UBO was obtained through a method described in our previous study [119]. Both oil samples (UBO and CNO) have a pH value of 4 [44].

### 4.3. Physico-Chemical Properties of Usnea barbata (L.) F.H. Wigg Extract in Canola Oil

#### 4.3.1. Atomic Force Microscopy

The equipment for Atomic Force Microscopy (AFM) was previously displayed [46]. Both oil samples (20 µL) were added to 2 mL of 96% ethanol (Merk Millipore, Burlington, MA, USA), further deposited on a clean glass substrate, and heated at 200 °C for 30 min. The characteristic line scans are displayed below the images in the so-called “enhanced contrast” mode [48], evidencing the surface profile of both oil samples.

#### 4.3.2. Rheology

Rheological tests were realized using a Kinexus Pro rheometer (Malvern Instruments Ltd., Malvern, Worcestershire, UK) equipped with a CF41 cryo-compact circulator (Julabo GmbH, Seelbach, Germany) and 50 mm plate-plate geometry with a set gap of 0.5 mm. All measurements were carried out at 25 °C. Amplitude tests were performed at a constant frequency of 1 Hz and 0.005–350 Pa; the results were plotted semi-logarithmically. Frequency sweep tests were recorded in the range of 0.1–50 Hz at constant shear stress, and the data were plotted logarithmically. The shear viscosity was measured at an applied shear rate between 10^−3^ and 10^3^ s^−1,^ and the flow curves were represented semi-logarithmically.

### 4.4. In Vitro Cytotoxicity

#### 4.4.1. Human Blood Cell Cultures

The blood cell cultures were obtained using the previously described technique [164]. After 72 h of incubation, the blood cell cultures were exposed to the samples and controls for 24 h, incubated in the same conditions [44].

#### 4.4.2. CLS-354 Cell Line, Cell Culture

As previously presented, the CLS-354 cells were cultured in their particular medium [166]. After dissociation with Trypsin-EDTA and centrifugation, the cancer cells were treated with samples and controls [44] and incubated for 24 h in the same conditions.

#### 4.4.3. Samples and Controls

The samples were O/W emulsions prepared with an oil phase concentration of 30% *w/w*; the emulsifier was Poloxamer 407 (P407) dissolved in water, in a concentration of 5% *w/w,* as previously mentioned [119]. Both emulsions (UBO and CNO, with pH = 5.5) in three dilutions (culture media/emulsion = 3:3, 3:2, and 3:1) [119] were used to treat human blood cells and oral cancer cells CLS-354. The cytotoxicity of samples was evaluated using one negative control and two positive ones. The negative control was represented by 1% DMSO. The first positive control was the emulsifier P407 (using a culture media/5%P407 ratio of 3:3). The second positive control was usnic acid of 125 µg/mL in 1% DMSO [44].

#### 4.4.4. Equipment

This study platform was the Attune™ Acoustic focusing cytometer (Thermo Fisher Scientific Inc., Waltham, MA, USA) [165]. The Attune™ cytometric software version (v.) 1.2.5 (Thermo Fisher Scientific Inc.,Waltham, MA, USA) collected and processed the flow cytometry data [46]. All flow cytometry measurements and fluorescent microscopy images were achieved using suitable kits (mentioned in Section 4.1) in the previously detailed conditions [44].

#### 4.4.5. Data Analysis

All analyses were achieved in triplicate; the results were registered as mean values ± standard deviation (SD). The results were presented as percent (%) for cell apoptosis, the enzymatic activity of caspase 3/7, autophagy, cell cycle, DNA synthesis, and count (×10^4^) of ROS in oxidative stress evaluation. After flow cytometry analyses, the data were processed using IBM^®^ SPSS^®^ Statistics v. 23.0 (IBM, New York, NY, USA). The Levene test examined the homogeneity of the sample’s variances. In addition, paired *t*-test and ANOVA established the differences between samples and controls, where *p* < 0.05 was statistically significant. Principal Component Analysis by XLSTAT v. 2022.2.1.1309 (Addinsoft, New York, NY, USA) investigated the correlations between all variable parameters [46].

### 4.5. Antibacterial and Antifungal Activities

The bacterial and fungal inocula preparation used the direct colony suspension method [167], as previously shown [48]. The samples were UBO and CNO in the emulsioned form. As the standard for bacteria, Ceftriaxone (Cefort 1g, provided by Antibiotice SA, Iasi, Romania) was used (as solutions of 30 mg/mL and 122 mg/mL prepared in distilled water). Terbinafine solution 10.1 mg/mL (Rompharm Company S.R.L., Otopeni, România) was selected as the standard for *Candida* sp. [47]. The solution of 5% P407 was used as a positive control [44]. The method used was a Resazurin-Based 96-Well Plate Microdilution Assay [168,169], previously detailed [46]. The colors from 96-well plates obtained after 24 h incubation were examined to highlight the differences between the standards and samples [101]. The sample concentrations were compared to the ones of the standard antibiotic. For yeasts, the results were interpreted using the resazurin color chart [100,170].

## 5. Conclusions

This work reveals the ROS-mediated anticancer potential of UBO through DNA damage (proved by high levels of nuclear shrinkage, cell cycle arrest in G0/G1, and DNA fragmentation) and autophagy. Moreover, through additional in vitro analyses, our study suggests that the synergism between lichen secondary metabolites and canola oil phytoconstituents could underly the UBO cytotoxicity.

Further steps could explore other pharmacological activities to evidence such synergic interaction. Moreover, future research could develop optimal pharmaceutical formulations with promising applications in oral cavity medicine.

## Figures and Tables

**Figure 1 ijms-23-14836-f001:**
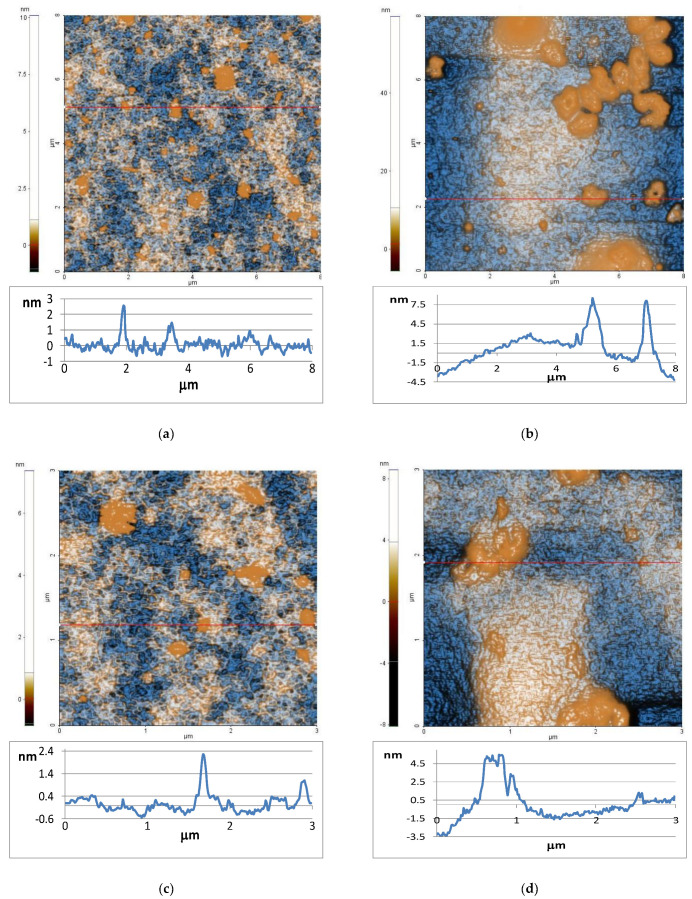
2D-AFM images and characteristic line scans (marked with horizontal red lines) at the scale of (8 × 8) μm^2^ and (3 × 3) μm^2^ for UBO (**a**,**c**) and CNO (**b**,**d**). Roughness (Rq) and peak-to-valley (Rpv) over the entire scanned areas: (8 × 8) μm^2^ (**e**) and (3 × 3) μm^2^ (**g**) and, respectively, along the line scans over 8 μm (**f**) and 3 μm (**h**).

**Figure 2 ijms-23-14836-f002:**
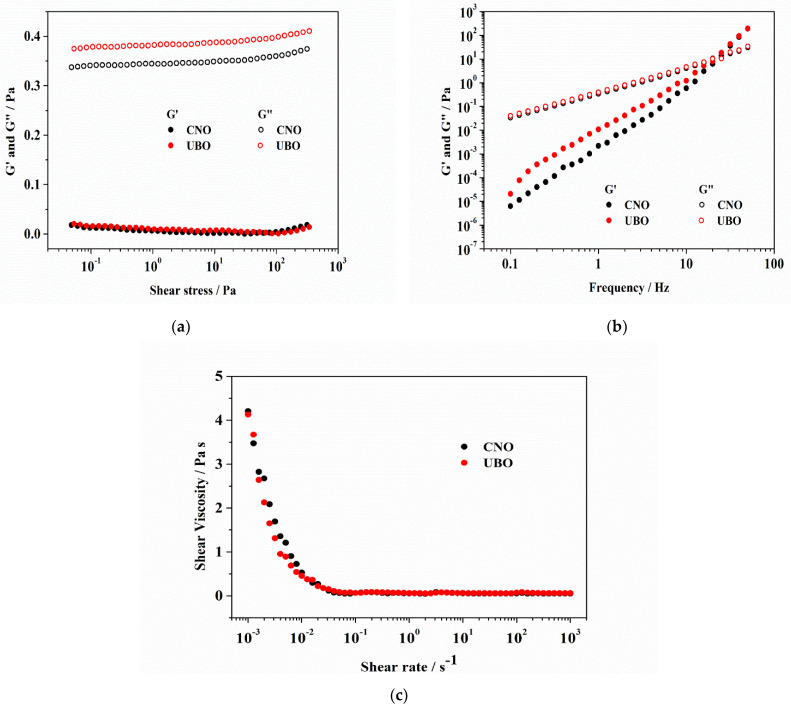
Stress sweeps for oil samples: CNO and UBO (**a**); frequency sweeps of CNO and UBO (**b**), where G’—storage modulus (represented as filled symbols) and G”—loss modulus (evidenced as empty symbols); Flow curves of CNO and UBO, at 25 °C (**c**).

**Figure 3 ijms-23-14836-f003:**
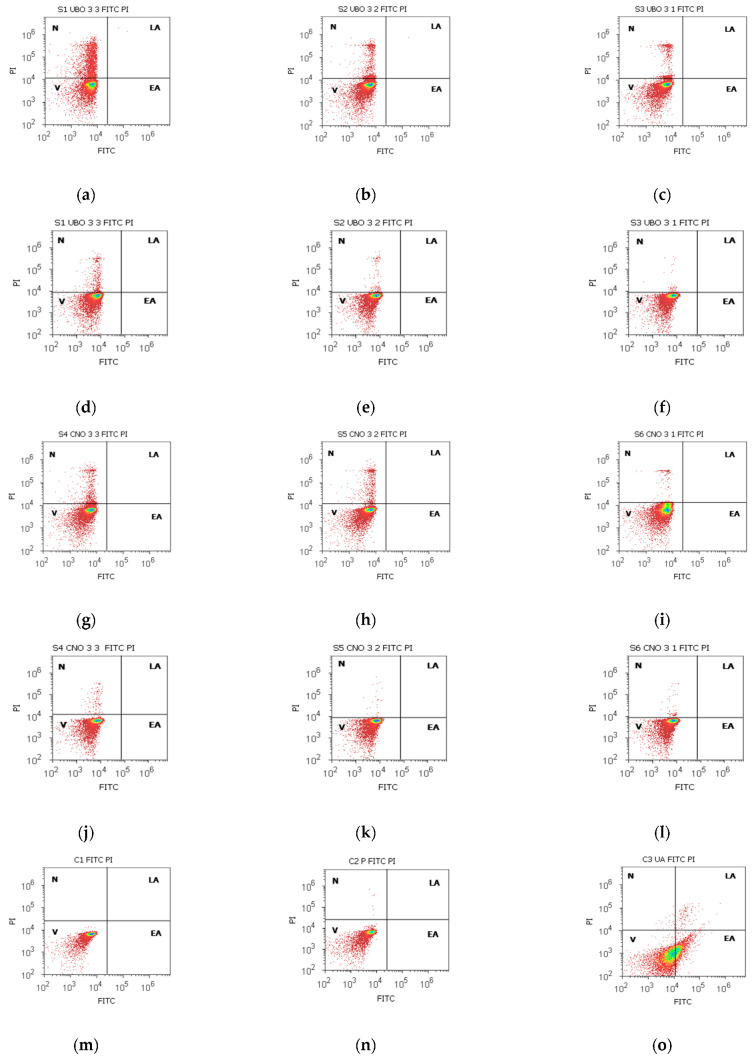
Apoptosis in blood cells (**a**–**c**,**g**–**i**,**m**–**o**) and CLS-354 tumor cells (**d**–**f**,**j**–**l**,**p**–**r**) after 24 h exposure to different concentrations [3:3; 3:2; 3:1] of *U. barbata* extract in canola oil (UBO) (**a**–**f**) and canola oil (CNO) (**g**–**l**) compared to C1, C2P, and C3UA controls (**m**–**r**). C1—1% DMSO (negative control), C2P—5% Poloxamer 407 (positive control), C3UA—usnic acid (positive control), V—viability, EA—early apoptosis, LA—late apoptosis, N—necrosis. [3:3; 3:2; 3:1]—culture medium/sample ratio (*v*/*v*); FM images of tumor cells stained with Annexin V-FITC/PI (**s**,**t**): viable (unstained) and early apoptotic cells—green stain (**s**); late apoptotic cells mixed with necrotic ones—green membrane with dark orange fragmented nuclei (**t**).

**Figure 4 ijms-23-14836-f004:**
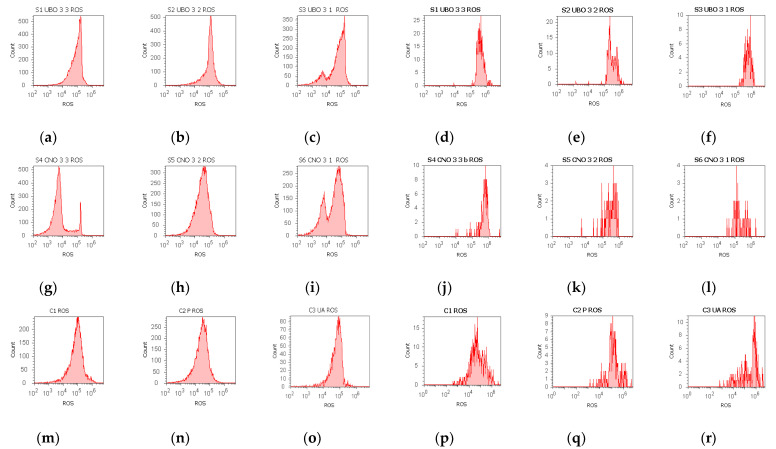
Total ROS activity in blood cells (**a**–**c**,**g**–**i**,**m**–**o**) and CLS-354 tumor cells (**d**–**f**,**j**–**l**,**p**–**r**) after 24 h exposure to different concentrations [3:3; 3:2; 3:1] of *U. barbata* extract in canola oil (UBO) (**a**–**f**) and canola oil (CNO) (**g**–**l**), compared to C1, C2P, and C3UA controls (**m**–**r**). C1—1% DMSO (negative control), C2P—5% Poloxamer 407 (positive control), C3UA—usnic acid (positive control), ROS—reactive oxygen species. [3:3; 3:2; 3:1]—culture medium/sample ratio (*v*/*v*).

**Figure 5 ijms-23-14836-f005:**
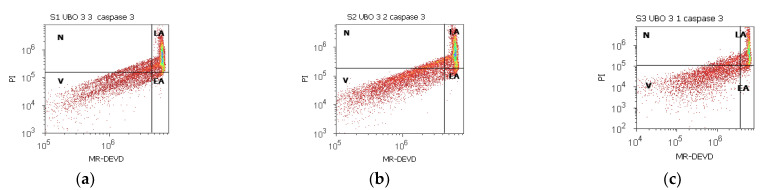
Enzymatic activity of caspase 3/7 in blood cells (**a**–**c**,**g**–**i**,**m**–**o**) and CLS-354 tumor cells (**d**–**f**,**j**–**l**,**p**–**r**) after 24 h exposure to different concentrations [3:3; 3:2; 3:1] of *U. barbata* extract in canola oil (UBO) (**a**–**f**) and canola oil (CNO) (**g**–**l**) compared to C1, C2P, and C3UA controls (**m**–**r**). C1—1% DMSO (negative control), C2P—5% Poloxamer 407 (positive control), C3UA—usnic acid (positive control); V—viability; EA—early apoptosis; LA—late apoptosis; N—necrosis. [3:3; 3:2; 3:1]—culture medium/sample ratio (*v*/*v*).

**Figure 6 ijms-23-14836-f006:**
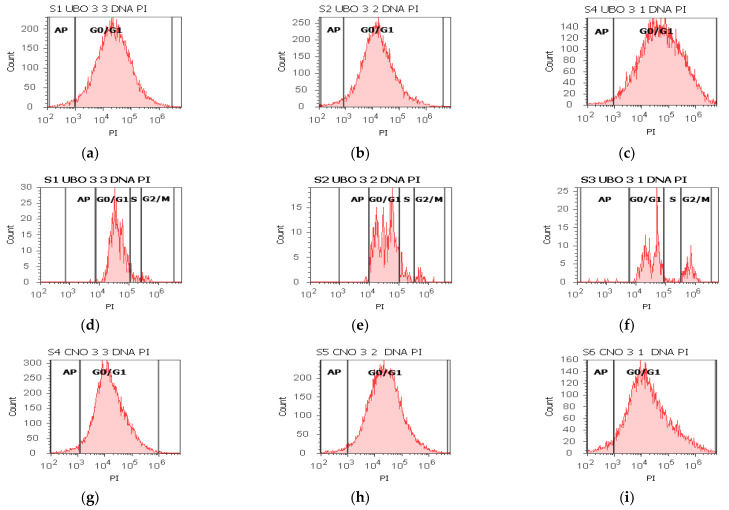
Cell cycle in blood cells (**a**–**c**,**g**–**i**,**m**–**o**) and CLS-354 tumor cells (**d**–**f**,**j**–**l**,**p**–**r**) after 24 h exposure to different concentrations [3:3; 3:2; 3:1] of *U. barbata* extract in canola oil (UBO) (**a**–**f**) and canola oil (CNO) (**g**–**l**) compared to C1, C2P, and C3UA controls (**m**–**r**). C1—1% DMSO (negative control), C2P—5% Poloxamer 407 (positive control), C3UA—usnic acid (positive control), AP—cell apoptosis (DNA fragmentation, subG0/G1), S—DNA synthesis; [3:3; 3:2; 3:1]—culture medium/sample ratio (*v*/*v*).

**Figure 7 ijms-23-14836-f007:**
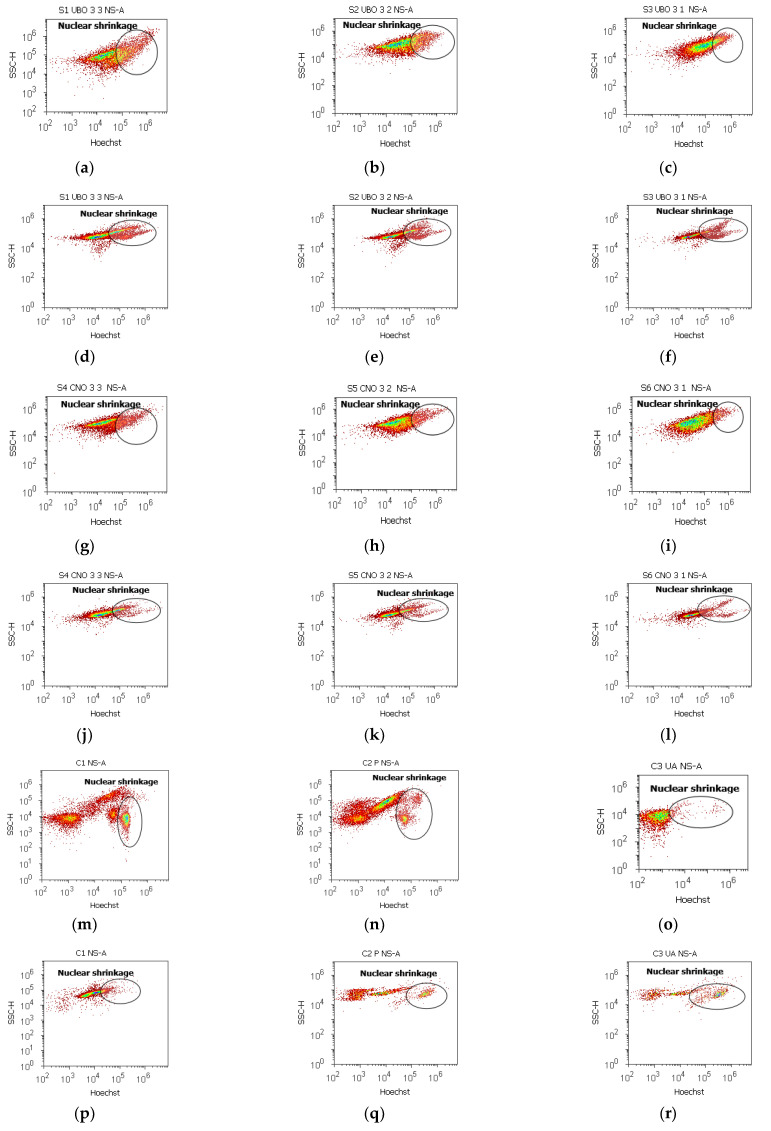
Nuclear shrinkage in blood cells (**a**–**c**,**g**–**i**,**m**–**o**) and CLS-354 tumor cells (**d**–**f**,**j**–**l**,**p**–**r**) after 24 h exposure to different concentrations [3:3; 3:2; 3:1] of *U. barbata* extract in canola oil (UBO) (**a**–**f**) and canola oil (CNO) (**g**–**l**) compared to C1, C2P, and C3UA controls (**m**–**r**). C1—1% DMSO (negative control), C2P—5% Poloxamer 407 (positive control), C3UA—usnic acid (positive control), NS—nuclear shrinkage, A—autophagy. [3:3; 3:2; 3:1]—culture medium/sample ratio (*v*/*v*).

**Figure 8 ijms-23-14836-f008:**
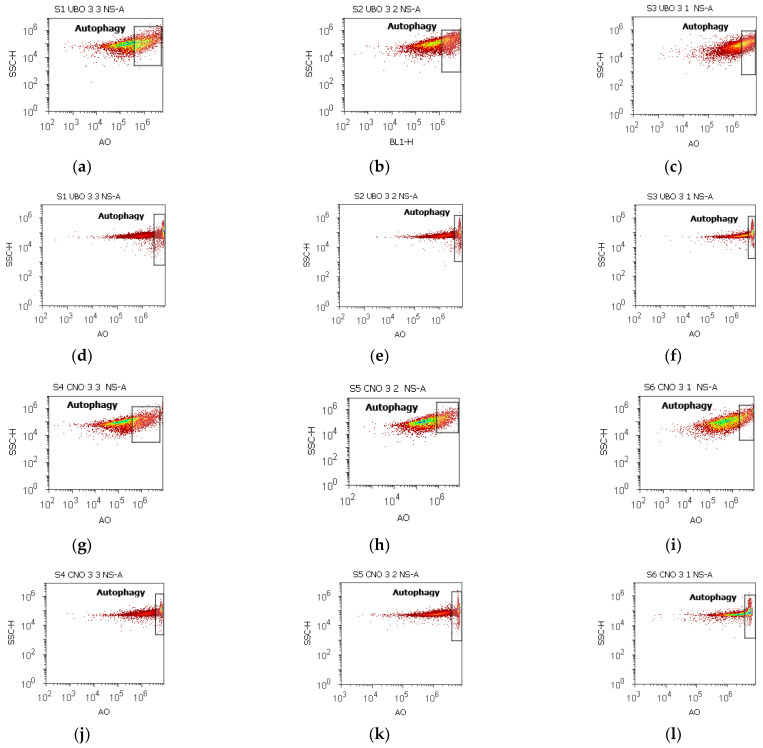
Autophagy in blood cells and CLS-354 tumor cells after 24 h exposure to different concentrations [3:3; 3:2; 3:1] of *U. barbata* extract in canola oil (UBO) (**a**–**f**) and canola oil (CNO) (**g**–**l**) compared to C1, C2P, and C3UA controls (**m**–**r**). C1—1% DMSO (negative control), C2P—5% Poloxamer 407 (positive control), C3UA—usnic acid (positive control), A—Autophagy, NS—Nuclear shrinkage, [3:3; 3:2; 3:1]—culture medium/sample ratio (*v*/*v*).

**Figure 9 ijms-23-14836-f009:**
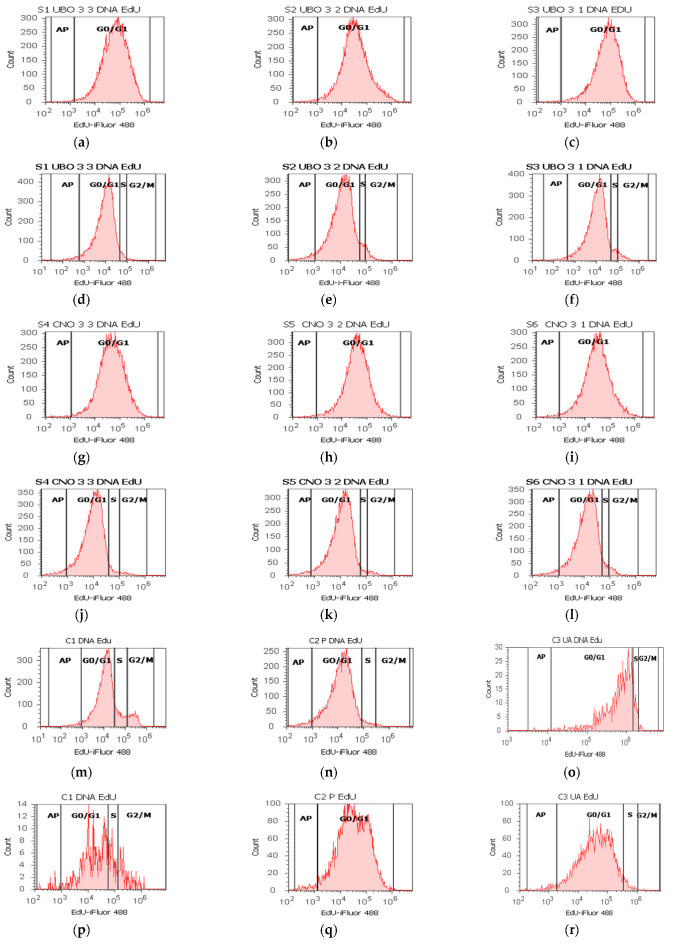
DNA synthesis and fragmentation in blood cells (**a**–**c**,**g**–**i**,**m**–**o**) and CLS-354 tumor cells (**d**–**f**,**j**–**l**,**p**–**r**) after 24 h exposure to different concentrations [3:3; 3:2; 3:1] of *U. barbata* extract in canola oil (UBO); (**a**–**f**) and canola oil (CNO) (**g**–**l**) compared to C1, C2P, and C3UA controls (**m**–**r**). C1—1% DMSO (negative control), C2P—5% Poloxamer 407 (positive control), C3UA—usnic acid (positive control), AP—cell apoptosis (DNA fragmentation, subG0/G1), S—DNA synthesis. [3:3; 3:2; 3:1]—culture medium/sample ratio (*v*/*v*).

**Figure 10 ijms-23-14836-f010:**
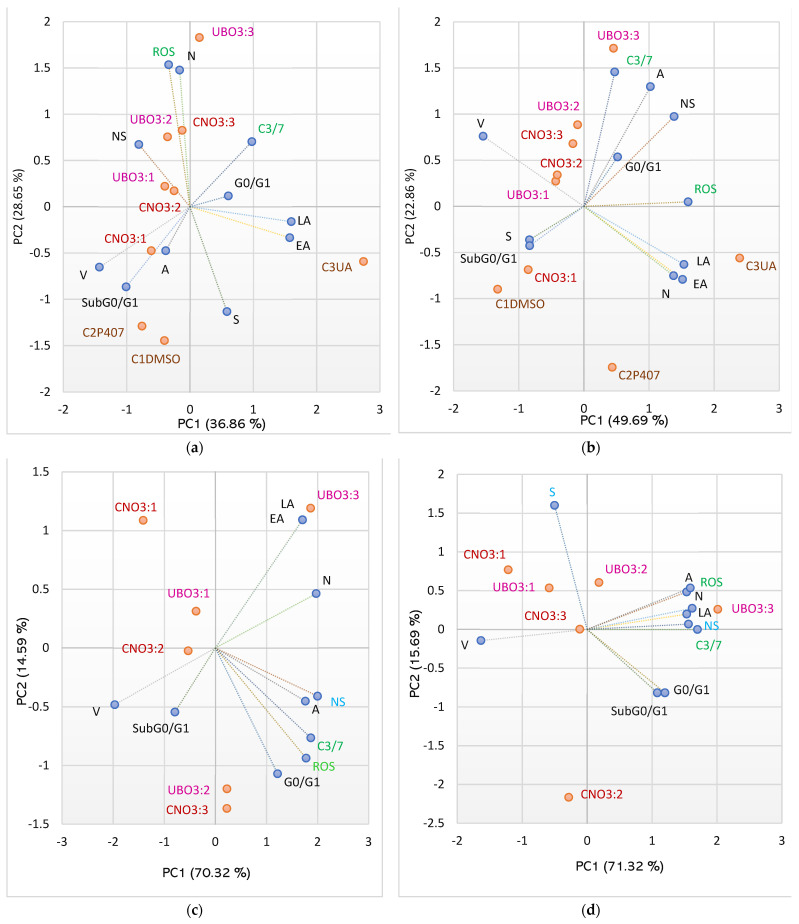
PCA-Correlation biplot (**a**,**b**) between mechanisms (enzymatic activity of caspase 3/7 and cellular oxidative stress) and processes induced by UBO, CNO, and controls (C1-DMSO, C2P407, and C2UA) in blood cells (**a**) and CLS-354 tumor cells (**b**). PCA-Correlation biplot (**c**,**d**) between mechanisms and induced processes only by oil samples (UBO and CNO) in blood cells (**c**) and CLS-354 cancer cells (**d**). UBO—*U. barbata* extract in canola oil, CNO—canola oil, [3:3, 3:2, 3:1] the ratio (*v*/*v*) between culture medium and oil sample. V—viability, EA—early apoptosis, LA—late apoptosis, N—necrosis, ROS—oxidative stress, C3/7—the enzymatic activity of caspase 3/7, A—autophagy, NS—nuclear shrinkage, DNAs—DNA synthesis, subG0/G1—apoptotic cell fraction, G0/G1—cell cycle arrest in G0/G1, C1—negative control with 1% dimethyl sulfoxide (DMSO), C2P—5% Poloxamer 407 (positive control), C3UA—usnic acid (positive control).

**Table 1 ijms-23-14836-t001:** Standard antibiotic (CTR), antifungal drug (TRF), Poloxamer 407 (P407), and oil samples (UBO and CNO) microdilutions. The inhibitory activity of oil samples (UBO and CNO) on bacterial species (*S. aureus* and *P. aeruginosa)* and *Candida* sp. (*C. albicans* and *C. parapsylosis*); the color score and signification [100].

CTR (mg/mL)	TRF (mg/mL)	P407 (mg/mL)	UBO/CNO (mg/mL)
30.230 ± 0.630	122.330 ± 0.850	10.050 ± 0.180	50.133 ± 1.305	303.300 ± 15.275
1.511 ± 0.043	6.117 ± 0.042	0.500 ± 0.009	2.506 ± 0.065	15.167 ± 0.764
0.755 ± 0.022	4.893 ± 0.034	0.250 ± 0.004	1.253 ± 0.032	7.583 ± 0.382
0.377 ± 0.011	3.914 ± 0.027	0.125 ± 0.002	0.626 ± 0.016	3.790 ± 0.193
0.188 ± 0.005	3.131 ± 0.021	0.061 ± 0.001	0.315 ± 0.008	1.895 ± 0.097
0.094 ± 0.002	2.505 ± 0.017	0.031 ± 0.001	0.157 ± 0.004	0.947 ± 0.048
0.047 ± 0.002	2.004 ± 0.014	0.015 ± 0.001	0.078 ± 0.002	0.473 ± 0.024
0.023 ± 0.001	1.603 ± 0.011	0.007 ± 0.001	0.039 ± 0.001	0.237 ± 0.012
** *S. aureus* **	** *P. aeruginosa* **
UBO	CNO	P407	CTR	UBO	CNO	P407	CTR
	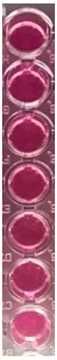		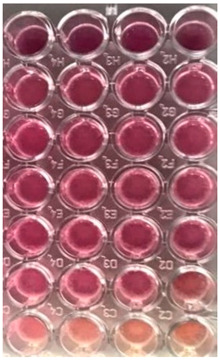	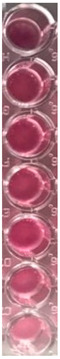	
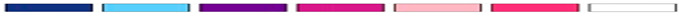 *
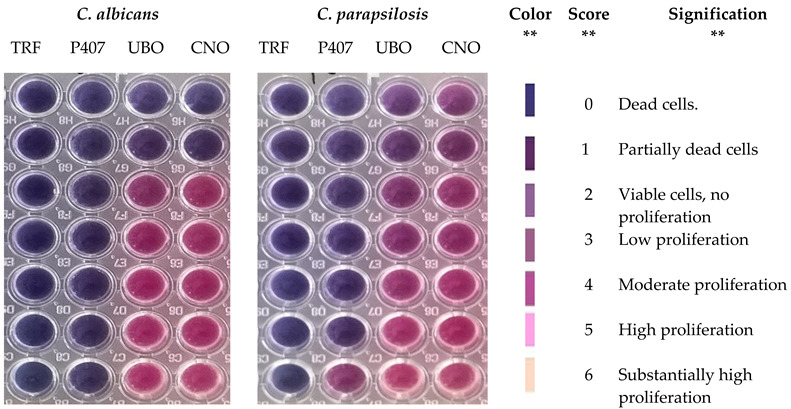

CNO—canola oil; UBO—*U. barbata* ethanol extract in canola oil; P407—Poloxamer 407 (positive control) CTR—Ceftriaxone; TRF—Terbinafine. * Well plates were examined using a resazurin dye chart adapted from Madushan et al. [101]. ** Results’ interpretation was adapted from Bitacura et al. [100].

## Data Availability

Not applicable.

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
