# Peer review of "ROS-Induced DNA-Damage and Autophagy in Oral Squamous Cell Carcinoma by Usnea barbata Oil Extract—An In Vitro Study"

_ijms, 2022, doi:10.3390/ijms232314836_

Round 1

Reviewer 1 Report

Overview and general recommendation:

In the manuscript, the authors explore the functions of U. barbata extract in canola oil (UBO). Flow cytometry is employed to show the effect of UBO on apoptosis, reactive oxygen species (ROS) levels, caspase 3/7 activity, cell cycle, nuclear shrinkage, autophagy, and synthesis of DNA. CLS-354-tumor cells and blood cells. They also tested the inhibitory activity of UBO against different pathogens.

I find the paper is organized in a proper way and the results are well described. The authors perform background research carefully. And major methods are well described in the manuscript and properly used in the research. All the results are good enough to support the conclusions. A lot of data is presented here, the authors should explain the inner logic that how they design the research and why they want to do these experiments. I also suggest the authors to reorganize the figures and the results, and present them in a more efficient way because the figures takes too much space of the paper.

Major comments:

1.      Figure1 g,h,i,j,k,l show the effect of  canola oil (CNO) in blood cells apoptosis(Figure1 g,h,i) and CLS-354 tumor cells apoptosis(Figure1 j,k,l). It seems that CNO itself will result in some apoptosis and the effect seems to be dose-dependent. I hope the authors can explain about this.

In Figure1 d,f,e, the UBO shows dose-dependent effect on cell viability of CLS-354-tumor cells. But it is confusing that in the column figure1 t,v,x, UBO [3:2] and UBO [3:1] don’t show any effect on cell viability of CLS-354-tumor cells. Can the authors explain why it seems different in two different figures?

2.      In Figure2, the authors claim that “UBO significantly stimulated ROS production compared to….”. The overall ROS level results from the ability of ROS production and ROS removal. The authors should show more evidence or reference if they want to claim that UBO can stimulate ROS production instead of inhibit ROS removal.

3.      I think the authors should explain the purpose of rheology research in this study. If you want to show the physical properties of UBO and CNO, I suggest the authors to put it in the beginning of the result.

4.      I suggest the authors to add an experiment the cell permeability of the UBO.

Reviewer 2 Report

The article  titled: “ROS-Induced DNA-Damage and Autophagy in Oral Squamous Cell Carcinoma by Usnea barbata Oil Extract an in Vitro Study” by Popovici et al., is an interesting manuscript but it contains some errors that need to be corrected before printing, so I recommend minor revision.

In my opinion, the abstract is too long and dispersive. Authors should make it shorter and more informative. They  should report a Background, highlighting  the purpose of the study and Methods. I would suggest to provide the most important Results of the study (including some quantitative values) and Conclusions. The abstract should be an objective representation of the article and it must not contain results that are not presented and substantiated in the main text and should not exaggerate the main conclusions.

Result section. The authors analyze autophagic processes through acridine orange labeling. In this way, the authors highlight the final phase of autophagy (acidic vesicular organelles) and not the initial and intermediate phase of the process (LC3 lipidation and autophagisome formation). Authors should clarify the reason for this choice and discuss it in the discussion section.

Round 2

Reviewer 1 Report

The authors test the effect of U. barbata extract in canola oil (UBO) on apoptosis, reactive oxygen species (ROS) levels, caspase 3/7 activity, cell cycle, nuclear shrinkage, autophagy, and synthesis of DNA by flow cytometry. I find that the authors have put considerable effort into addressing the reports of the referees. They add more information in the background and explain that the CNO itself will affect apoptosis. And they make proper changes in organizing the results and figures. As a result the paper is very much improved and I have no problem in recommending it for publication.